# Novel 3,19-(N-Phenyl-3-(4-fluorophenyl)-pyrazole) Acetal of Andrographolide Promotes Cell Cycle Arrest and Apoptosis in MDA-MB-231 Breast Cancer Cells

**DOI:** 10.3390/ph18071026

**Published:** 2025-07-10

**Authors:** Siva Kumar Rokkam, Shahjalal Chowdhury, Yashwanth Inabathina, Lakshminath Sripada, Srinivas Nanduri, Balasubramanyam Karanam, Nageswara Rao Golakoti

**Affiliations:** 1Department of Chemistry, Sri Sathya Sai Institute of Higher Learning, Puttaparthi 515134, Andhra Pradesh, India; rokkamsivakumar8877@gmail.com (S.K.R.); sripadalakshminath@sssihl.edu.in (L.S.); 2Department of Biology and Cancer Research, Tuskegee University, Tuskegee, AL 36088, USA; schowdhury8950@tuskegee.edu (S.C.); yinabathina8685@tuskegee.edu (Y.I.); 3Department of Chemical Sciences, National Institute of Pharmaceutical Education and Research, Balanagar, Hyderabad 500037, Telangana, India; nandurisrini92@gmail.com

**Keywords:** pyrazole acetals, andrographolide, isoandrographolide, anticancer, antioxidant

## Abstract

**Background:** Natural products play a crucial role in cancer treatment due to their ability to selectively target cancer cells. Andrographolide, a major constituent of *Andrographis paniculata*, exhibits potential anticancer properties. Considering the pharmacological importance of nitrogen-based heteroaromatic scaffolds, particularly pyrazole motifs, this study aimed to integrate the pyrazole pharmacophore with the andrographolide scaffold to develop novel therapeutic candidates. **Methods:** Twenty novel 3,19-(N-phenyl-3-aryl-pyrazole) acetals of andrographolide and isoandrographolide were synthesized and characterized using UV-Vis, FT-IR, NMR, and HRMS. Initial anticancer screening was conducted by the National Cancer Institute (NCI), USA, against 60 human cancer cell lines. The most promising compound, **1f** (R = 4-F), was selected for further biological evaluation in the MDA-MB-231 breast cancer cell line. **Results:** The MTT assay results demonstrated that compound **1f** exhibited strong, dose-dependent anti-proliferative effects. The apoptosis analysis of **1f** revealed a time-dependent increase in apoptotic cells, and cell cycle studies indicated S phase arrest in MDA-MB-231 cells. Antioxidant activity via the DPPH assay identified compounds **1b** (R = 3-NO_2_) and **2b** (R = 3-NO_2_) as the most effective radical scavengers. The most active compounds were also evaluated for drug-likeness using in silico Lipinski’s rule assessments. **Conclusions:** The synthesized 3,19-(N-phenyl-3-aryl-pyrazole) acetals of andrographolide and isoandrographolide exhibited promising anticancer and antioxidant properties. Among them, compound **1f** showed the most significant activity, supporting its potential as a lead candidate for further anticancer drug development.

## 1. Introduction

Cancer is a group of diseases that occur when cells divide uncontrollably and consistently [1]. Despite the achievements of oncology, cancer continues to be one of the most fatal diseases and the second leading cause of mortality in the world [2]. As stated by the GLOBOCAN 2020 global cancer statistical report, ~19.3 million new cancer cases were recorded and accounted for ~10 million deaths worldwide (barring non-melanoma skin cancer) [3]. Breast, lung, colon, prostate, skin, and stomach cancer are the most common cancer, and their fatality rate is considerably high [4]. The most prevalent cancer diagnosed currently is breast cancer, with an estimated 2 million new cases annually, and remains the 5th highest cause of cancer death, with over half a million deaths [5,6]. Approximately 28 million new cases of cancer are anticipated worldwide over the next twenty years, a 47% increase from 2020 [3,7]. Current anticancer drugs used in treatment often cause adverse effects such as weakened immune function, hair loss, cardiotoxicity, and infertility in women with active ovaries [8,9,10]. Consequently, alternative treatments involving natural products, which offer greater potential with fewer side effects for cancer patients, should be prioritized [11,12,13,14].

Natural products have been an excellent primary source of bioactive molecules. These bioactive molecules have acquired great value in cancer drug discovery due to their structural features and wide range of medicinal properties [15,16,17,18]. One such bioactive molecule is andrographolide (ADG). It is an active diterpenoidal constituent of *Andrographis paniculata* Nees (Figure 1), which belongs to the family Acanthaceae [19,20]. It is primarily accumulated in the leaves, while modest amounts can also be found in the flowering tops, stems, and roots [21]. It is considered an easily isolable compound with a high yield. *Andrographis paniculata*, also commonly known as the ‘King of Bitter’, is widely cultivated in tropical and subtropical Southeast Asia, China, and India [22]. In traditional Chinese and Indian medicine, the leaves and roots of this plant were used as a remedy for various ailments such as fever, common cold, mouth ulcers, skin infections, urinary tract infections, gastrointestinal disorders, to dispel the toxins from the body, venomous snakebites, etc. [23,24,25]. Several research groups investigated the extracts and compounds isolated from *A. paniculata* and reported that they displayed a wide range of pharmacological activities, including anticancer, antiviral, anti-inflammatory, antimalarial, antioxidant, immunostimulatory activities, etc. In addition, these investigations revealed that the observed anticancer properties of the methanolic extracts of the plant are mainly attributed to andrographolide [26,27,28,29,30,31,32,33,34].

In various human cancer cell lines, andrographolide was evaluated for its anticancer potential both in vitro and in vivo [35]. The studies disclosed that andrographolide targets several signaling pathways and holds significant promise for chemoprevention in various types of cancer. By interacting with a variety of targets, andrographolide affects several cancer-related characteristics, such as an increase in the expression of pro-apoptotic genes that lead to apoptosis by either intrinsic mitochondrial death or extrinsic TRAIL-related death, cell cycle arrest at different stages with the enhancement of pro-apoptotic gene and protein activities, anti-angiogenic properties, the inhibition of the VEGF pathway, NF- κβ inhibition, the ubiquitin-mediated proteasomal degradation of proteins, and the inhibition of IL-mediated signals [35,36,37,38,39]. Though it exhibited excellent therapeutic potential, it has limited oral bioavailability due to its quick elimination and high plasma protein-binding capacity [40]. This makes creating formulations for clinical treatment extremely difficult. However, owing to its unique structural features, several structural changes have been made to various parts of andrographolide to enhance its anticancer activity and oral bioavailability. Interestingly, these modifications resulted in a significant increment in the anticancer potential and bioavailability. Like andrographolide, pyrazole derivatives also exhibit remarkable anticancer effects via the inhibition of various kinds of enzymes, proteins, and receptors, such as CDKs, kinases, EGFR, VEGF, TGF-β, etc. [41,42,43,44,45].

In this context, we have synthesized twenty novel 3,19-(N-phenyl-3-aryl-pyrazole) acetals of andrographolide and isoandrographolide. Our main intention is to link the pyrazole scaffold to andrographolide due to its widespread pharmacological effects in medicinal chemistry, mainly as an anticancer and anti-inflammatory pharmacophore. As expected, the synthesized derivatives have displayed excellent anticancer activity, several-fold increased activity compared to andrographolide, against the breast cancer cell line MDA-MB-231.

## 2. Results and Discussion

### 2.1. Chemistry

The synthesis of 3,19-(N-phenyl-3-aryl-pyrazole) acetals of andrographolide and isoandrographolide entails a series of consecutive steps. This synthesis encompasses three distinct steps, and the schematics detailing the adopted synthetic strategies are presented in Figure 1, Figure 2 and Figure 3 [46]. In the initial step, ten 1-phenyl-2-(1-arylethylidene) hydrazines (**A**–**J**) were synthesized. Then, in the second step, these compounds were transformed into the corresponding N-aryl-pyrazole-4-carboxaldehydes (**a**–**j**). Lastly, in the third step, these aldehydes were subjected to reactions with andrographolide (ADG) and isoandrographolide (ISOADG), resulting in the final attainment of 3,19-(N-phenyl-3-aryl-pyrazole) acetals of andrographolide and isoandrographolide (**1a**–**1j** and **2a**–**2j**).

### 2.2. Characterization

The UV-Vis spectra exhibited bands between 275 and 280 nm for all the synthesized compounds (**1a**–**1j** and **2a**–**2j**). In the IR spectra of 3,19-(N-phenyl-3-aryl-pyrazole) acetals of andrographolide (**1a**–**1j**), peaks at ~3400 cm^−1^ and ~1672 cm^−1^ were observed for -OH and exocyclic C=C groups. These peaks were absent for 3,19-(N-phenyl-3-aryl-pyrazole) acetals of isoandrographolide (**2a**–**2j**). The presence of the C-H stretching band within the 3089–3070 cm^−1^ range and the aromatic skeletal bands spanning from 1600 to 1450 cm^−1^ confirms the existence of the aromatic group. The band situated at 2948–2920 cm^−1^ corresponds to the sp^3^ C-H stretch originating from the diterpene moiety. The peak in the range of 1732–1759 cm^−1^ corresponds to the carbonyl group (C=O) of the butyrolactone ring. Distinct absorptions at approximately 1220 and 1100 cm^−1^ can be attributed to the C-O stretching. Compounds with aryl chloride have a band at 1071–1060 cm^−1^ (Ar-Cl stretch). Compounds with aryl fluorides have a band around 1245 and 1220 cm^−1^ (Ar-F stretch), and compounds with aryl bromides have a band between 1020 and 1011 cm^−1^ (Ar-Br stretch).

The attachment of the substituted pyrazole moiety is discernible in ^1^H NMR and ^13^C NMR spectra of all the acetals. In ^1^H NMR, the peak at ~δ 8.18 is due to the presence of only hydrogen in the pyrazole ring, whereas peaks around δ 8.76 to 6.99 correspond to the protons on the aromatic ring attached to the pyrazole ring. In ^13^C NMR, the carbons in the pyrazole ring and aromatic ring were identified in the ~δ 163 to 115 range. The bridged carbon, which was bonded to two oxygen atoms, provides strong evidence for the formation of the acetals, and its peak was observed at δ 90 in the ^13^C NMR spectrum, and the proton attached to the acetal carbon is observed at ~δ 5.87 in the ^1^H NMR. However, the carbonyl group (C=O) signals in the acetals appeared approximately within the range of δ 173–170. In 3,19-(N-phenyl-3-aryl-pyrazole) acetals of andrographolide, peaks at ~148 and 110 in the ^13^C NMR spectra correspond to an exocyclic C=C, which is absent in 3,19-(N-phenyl-3-aryl-pyrazole) acetals of isoandrographolide [47,48]. More detailed information on the characterization data is provided in the Appendix A.

### 2.3. In Vitro Anticancer Studies

#### 2.3.1. NCI Screening

All the synthesized compounds (**1a**–**1j** and **2a**–**2j**) were initially assessed for their anticancer potential at the National Cancer Institute (NCI, USA) against a diverse panel of 60 human cancer cell lines, representing nine distinct cancer types including leukemia, non-small-cell lung, colon, CNS, melanoma, ovarian, renal, prostate, and breast. The testing protocol employed by the NCI encompasses two stages: initial assessment at a single dose and subsequent evaluation at five doses. Compounds that demonstrated growth inhibition exceeding 32% at the single-dose stage were identified as active candidates (32% cutoff used by the NCI as a general guide to filter out inactive compounds).

#### 2.3.2. Single-Dose Study

Initially, a comprehensive anticancer screening of all twenty compounds (**1a**–**1j** and **2a**–**2j**), including the parent compounds ADG and ISOADG, was conducted using a 10 µM dose across all 60 cell lines (a standardized protocol followed for the initial assessment of test compounds by the NCI). In this assessment, a subset of ten compounds (**1a**–**1j**) demonstrated impressive percent growth inhibition in most of the cell lines. As a result, these ten compounds were chosen for five-dose studies. Of particular interest is the observation that all ten compounds exhibiting notable activity across most of the cell lines are 3,19-(N-phenyl-3-aryl-pyrazole) acetals of andrographolide (**1a**–**1j**). Conversely, 3,19-(N-phenyl-3-aryl-pyrazole) acetals of isoandrographolide (**2a**–**2j**), ADG, and ISODG displayed a lower % growth inhibition, suggesting that these derivatives are less potent across the majority of cell lines. Consequently, compounds **2a**–**2j**, ADG, and ISODG did not meet the criteria for further advancement in the study.

#### 2.3.3. Five-Dose Study

In the five-dose study, 3,19-(N-phenyl-3-aryl-pyrazole) acetals of andrographolide (**1a**–**1j**) were tested at concentrations of 0.01 µM, 0.1 µM, 1 µM, 10 µM, and 100 µM against 60 cancer cell line panels. The growth inhibition 50% (GI_50_), total growth inhibition (TGI), and lethal concentration 50% (LC_50_) of the ten potent compounds on the cancer cell lines are given in Table 1.

It is interesting to observe that all ten compounds have exhibited excellent GI_50_ values across all the cell lines. However, among these compounds, **1f** demonstrated the best activity, followed by **1j** and **1h**. Compounds **1i** and **1b** were slightly less active as compared to other compounds. In addition, we note that *para*-substituted derivatives (**1f**, **1g**, **1h**, **1i,** and **1j**) exhibited better GI_50_ values than *meta*-substituted derivatives (**1b**, **1c**, **1d,** and **1e**). The observed anticancer activity order of these compounds is as follows: **1f** > **1j** > **1h** > **1g** > **1d** > **1c** > **1e** > **1a** > **1b** > **1i**.

#### 2.3.4. GI_50_ Values of Compounds on Individual Cancer Types

Leukemia Cancer: The compound **1f** displayed the most potent activity on most of the leukemia cancer cell lines. However, the best GI_50_ (0.40 µM) shown by **1f** was against the cell line CCRF-CEM, followed by the compounds **1h** (GI_50_: 0.48 µM)**, 1a** (GI_50_: 0.59 µM), **1j** (GI_50_: 0.64 µM), **1e** (GI_50_: 0.70 µM), and **1b** (GI_50_: 1.14 µM). However, the compounds **1c**, **1d**, **1g**, and **1i** showed better activity against the cell line MOLT-4.

Non-Small-Cell Lung Cancer: The compound **1g** was observed to have the best GI_50_ (1.02 µM) against the NCI-H322M cell line. While the compounds **1a**, **1f**, **1i**, and **1j** were active against the HOP-92 cell line, compounds **1b**, **1c**, **1d**, and **1e** were active against the NCI-H522 cell line.

Colon Cancer: All the compounds except **1b**, **1c**, and **1i** were found to be active against the HCT-116 cell line. Compound **1f** demonstrated the best activity with GI_50_: 0.39 µM. Compounds **1b**, **1c**, and **1i** showed the best GI_50_ against HCT-15 and HCC-2998 cell lines.

CNS Cancer: Among the CNS cancer cell lines, these compounds showed promising effects against three cell lines. All the *para*-substituted derivatives (**1f**, **1g**, **1h,** and **1i**) except **1j** displayed excellent growth inhibition against the SNB-19 cell line, whereas **1j, 1a,** and **1d** are more active against SNB-75, and **1d** (GI_50_: 1.12 µM) is the best among all. On the other hand, *meta*-substituted derivatives (**1b**, **1c,** and **1e**) showed the best GI_50_s against SF-295.

Melanoma Cancer: As observed in CNS cancer, against the melanoma cancer cell line LOX IMVI, the *para*-substituted derivatives (**1f**, **1g**, **1h,** and **1i**) demonstrated outstanding growth inhibition with GI_50_s of 1.32, 1.58, 1.55, and 1.43 µM, respectively. All other compounds displayed better activity on various melanoma cancer cell lines.

Ovarian Cancer: These compounds were observed to have very good activity against different ovarian cancer cell lines. However, among all, compound **1j** (GI_50_: 1.35 µM) was found to be the most potent against OVCAR-8, followed by **1d** (GI_50_: 1.39 µM) against the IGROV1 cell line.

Renal Cancer: As in the case of CNS and melanoma cancer, *para*-substituted derivatives (**1f**, **1g**, **1h,** and **1i**), except **1j,** exhibited the best activity with GI_50_s of 1.27, 1.33, 1.19, and 1.33 µM against TK-10. However, compound **1j** (GI_50_: 0.89 µM) was the most active against RXF 393 amongst all renal cancer cell lines. The *meta*-substituted derivatives (**1b**, **1d**, and **1e**), except **1c** (active against TK-10, GI_50_: 1.41 µM), displayed the best activity against RXF 393 with GI_50_s of 1.62, 1.05, and 1.18 µM.

Prostate Cancer: Between the two types of cancer cell lines (PC-3 and DU-145) against which the compounds were tested, four compounds **1a**, **1b**, **1f**, and **1i** showed the best GI_50_s against PC-3, while remaining six compounds **1c**, **1d**, **1e**, **1g**, **1h**, and **1j** showed the best GI_50_s against DU-145.

Breast Cancer: Including **1a** and **1b** (GI_50_: 1.42 and 1.76 µM), the *para*-substituted derivatives (**1f** (GI_50_: 1.19 µM), **1g** (GI_50_: 1.39 µM), **1h** (GI_50_: 1.18 µM), and **1i** (GI_50_: 1.14 µM)) also demonstrated the highest growth inhibition against the T-47D cell line. However, the compounds **1j** and **1d** (GI_50_: 1.46 and 1.48 µM) against MCF-7 and **1c** and **1e** (GI_50_: 1.30 and 1.46 µM) against BT-549 were more active.

#### 2.3.5. NCI Compare Analysis

The NCI compare analysis assesses and arranges compounds based on their similarity in response to the NCI 60 cell lines compared to marketed drugs [49]. This similarity is quantified using the Pearson Correlation Coefficient (PCC). When a marketed drug ranks high in this comparison alongside the compound being studied, it suggests that the studied compound might share a similar mechanism of action. The most promising compounds **1f**, **1h**, and **1j** from the five-dose study were chosen for NCI compare analysis, and their GI_50_ values were compared to those of marketed drugs (Table 2).

The NCI compare analysis revealed that the three compounds exhibited correlations with various marketed drugs that are recognized for their ability to inhibit DNA synthesis and promote apoptosis. This observation strongly suggests that these compounds could potentially impede cancer growth through similar mechanisms.

#### 2.3.6. Effect of **1f** on Proliferation of MCF-10a and MDA-MB-231

As compound **1f** exhibited the best anticancer activity among all the screened compounds in the NCI study, its cytotoxicity on breast cells MCF-10a (normal breast cells) was examined. Compound **1f** was found to have a low inhibitory concentration with a CC_50_ of 50.01 µM against the breast cells MCF-10a. Further, the MTT assay was performed for compound **1f** against the MDA-MB-231 cell line to check its cytotoxicity. These MDA-MB-231 breast cancer cells were treated with **1f** at different concentrations, i.e., 0, 1.56, 3.125, 6.25, 12.5, 25, 50, 100, and 200 µM, for 72 h. As shown in Figure 2, the absorbance decreased with the increasing concentration of the drug. The results indicate that **1f** has the strong dose-dependent anti-proliferation activity against the MDA-MB-231 breast cancer cell line.

#### 2.3.7. Effect of **1f** on Cell Apoptosis Inducement

Apoptosis is considered a preventive mechanism against cancer dissemination, as it plays a crucial role in eliminating excessively proliferating and mutated cancer cells from the body [50,51]. Following the administration of cytotoxic drugs as treatment for various types of cancer, apoptosis is a significant contributor to cellular death [52]. Hence, an enhanced understanding of the processes and mechanisms underlying apoptosis can aid in the development of novel therapeutic drugs for cancer [53]. We proceeded to investigate whether compound **1f** triggers apoptosis or not. The MDA-MB-231 breast cancer cell line was treated with **1f** for 24, 48, and 72 h. A quantitative evaluation of apoptotic activity was performed using FACS.

As depicted in Figure 3, a notable rise in apoptotic cells was noticed in a time-dependent manner. We observed 28.09% apoptotic cells in the untreated control (8.89% in early apoptosis and 19.2% in late apoptosis). Upon treatment with **1f**, while after 24 h apoptotic cells remained almost the same, we observed a gradual increase in apoptotic cells by 44% at 48 h (10.3% in early apoptosis and 33.0% at late apoptosis) and 55% at 72 h (14.7% in early apoptosis and 40.6% at late apoptosis), respectively, compared to the control at 0 h. Thus, confirming that compound **1f** promoted apoptosis in MDA-MB-231 breast cancer cells.

#### 2.3.8. Effects of **1f** on MDA-MB-231 Cell Cycle Distribution

Cell proliferation is well associated with the regulation of cell cycle progression. Therefore, the effect of **1f** on cell cycle distribution was evaluated by flow cytometry in PI-stained cells. As shown in Figure 4, after treatment for 24 h with **1f** at concentrations of 3.33, 5.0, and 6.66 µM, cells in G2/M distributions decreased from 18.9% to 4.01% compared to control cells (at 0 h), whereas the percentage of cells in the G1 phase decreased from 71.2% to 59.6% and cells in the S phase increased gradually from 8.51% to 14.2%, 25.2%, and 23.3%, respectively (Table 3). The result indicates that **1f** could induce cell cycle arrest in the S phase.

### 2.4. Antioxidant Activity

The onset and advancement of cancer have been associated with oxidative stress due to its role in escalating DNA mutations, triggering DNA impairment, fostering genome instability, and promoting cellular proliferation [54,55,56]. Therefore, to find out the antioxidant properties of the compounds, DPPH radical scavenging activity was performed. As shown in Figure 5, all the synthesized compounds showed a DPPH radical scavenging activity between 56.06% to 61.17% as compared to ADG (58.74%), ISOADG (57.94%), and the positive control ascorbic acid (AA, 69.26%). All the compounds demonstrated more or less similar antioxidant potential compared to ADG and ISOADG. However, the compounds with the strongest electron-withdrawing groups, i.e., **1b** (R = 3-NO_2_, 60.68%) and **2b** (R = 3-NO_2_, 61.17%), are the most active, whereas the compounds **1i** (R = 4-CH_3_, 56.31%) and **2i** (R = 4-CH_3_, 56.06 %) are the least active. Among the halogen-substituted derivatives, chloro-substituted compounds in both the series (**1d**, 59.32% and **2d**, 59.81%) have shown superior activity. Another interesting observation was that all the *meta*-substituted derivatives displayed slightly better radical scavenging activity than all the *para*-substituted derivatives in both series, except in the case of **2c**.

### 2.5. Drug-Likeness Studies for Active Compounds

Using the pkCSM tool, a computer-based assessment was conducted on the active compounds (**1a**–**1j**) to assess their suitability as drug candidates by examining their Lipinski parameters [57].

The analysis of the results, as given in Table 4, revealed that in silico molecular properties of the compounds align with Lipinski’s rule of five (RO5), except for their molecular weights, which fall within the range of 580–658 Daltons, slightly surpassing the recommended limit (<500 Daltons). Nevertheless, it is worth noting that natural products often deviate from Lipinski’s rules regarding molecular weights [58]. The compounds exhibited a TPSA ranging from 78 to 126, whereas the Log *p*-value slightly exceeded the acceptable range for some compounds. Additionally, they have fewer than five N-H and O-H hydrogen bond donors and fewer than ten nitrogen and oxygen hydrogen bond acceptors (HBA). The parent compound ADG shows better predicted bioavailability than its derivatives, due to its lower molecular weight, LogP, and balanced polarity. However, some of the derivatives showed better anticancer activity than ADG. Overall, despite the slight deviation in molecular weights, the compounds largely adhere to Lipinski’s parameters, suggesting their suitability as potential anticancer agents.

## 3. Experimental

### 3.1. Synthesis

All the chemicals used in the experiments were procured from Merck and S.D. Fine chemicals (Bangalore, India). Andrographolide (98%) was obtained from Maysar Herbals (Faridabad, Haryana, India) and purified by recrystallization. Solvents such as methanol and dichloromethane were distilled before use.

#### 3.1.1. Synthesis of Arylhydrazones (**A–J**)

To a solution of acetophenone (25 mmol) in methanol (80 mL), phenylhydrazine (25 mmol) was added and stirred at room temperature. To this, 2 mL acetic acid was added, and the mixture was refluxed for 24 h. The progress of the reaction was checked using thin-layer chromatography (TLC). After the completion of the reaction, crushed ice was added, and the solid formed was filtered and dried. The pure product obtained was then subjected to the next reaction [46].

#### 3.1.2. Synthesis of 3-Aryl-4-pyrazole carboxaldehydes (**a–j**)

To a cold solution of N, N-dimethylformamide (40 mL), aliquot amounts of arylhydrazone (20 mmol) were added and stirred over 30 min. Then, POCl_3_ (100 mmol) was added dropwise to the solution while maintaining the temperature at 0–5 °C. After the addition, the reaction mixture was heated at 55 °C for 4 h. After the completion of the reaction, the resulting mixture was cooled to room temperature and poured into crushed ice and stirred for 20 min. The white solid formed was filtered, dried, and purified using column chromatography (hexane/ethyl acetate, 75:25). In the case the solid was not formed after being poured into crushed ice, the solution was neutralized with saturated Na_2_CO_3_ to obtain the solid [46].

#### 3.1.3. Synthesis of 3,19-(N-phenyl-3-aryl-pyrazole) Acetals of Andrographolide (**1a**–**1j**)

Andrographolide (1 mmol) in DMSO (2 mL) was added to 3-aryl-4-pyrazole carboxaldehydes (4 mmol) in toluene (3 mL) at room temperature. To this catalytic amount (0.08 mmol) of pyridinium *para*-toluene sulfonate (PPTs) were added and heated at 70 °C for 24 h. After the completion of the reaction, the mixture was cooled to room temperature and then neutralized with a freshly prepared, cold saturated sodium bicarbonate solution. The resulting solution was then extracted three times (15 mL each) with DCM, and the obtained organic layer was washed thoroughly with brine solution and dried over anhydrous sodium sulphate. DCM was removed using a rota evaporator. The crude product obtained was then purified using column chromatography (silica gel: 230–400 mesh, hexane/acetone, 80:20) [46].

#### 3.1.4. Synthesis of Isoandrographolide and 3,19-(N-phenyl-3-aryl-pyrazole) Acetals of Isoandrographolide (**2a–2j**)

To synthesize isoandrographolide, andrographolide (8 g) was dissolved in concentrated hydrochloric acid (HCl, 150 mL) and stirred at room temperature for 24 h. After the completion of the reaction (monitored by TLC), the reaction mixture was poured into crushed ice and extracted with DCM. Then, the organic layer was thoroughly washed with brine and freshly prepared saturated sodium bicarbonate solution. The crude solid was then dried over anhydrous sodium sulphate and recrystallized from ethyl acetate to obtain a pure product [46].

3,19-(N-phenyl-3-aryl-pyrazole) acetals of isoandrographolide were synthesized as described in the synthesis of 3,19-(N-phenyl-3-aryl-pyrazole) acetals of andrographolide (in this case, isoandrographolide was used as a starting material instead of andrographolide).

### 3.2. Instrumentation

The ^1^H NMR and ^13^C NMR spectra were recorded using a Bruker Ascend NMR spectrometer (Bruker BioSpin AG, Faellanden, Switzerland) at 400 MHz and 100 MHz, respectively. CDCl_3_ was used as the solvent to dissolve the compounds. TMS was an internal standard. Agilent 6550 Q-TOF LC/MS (HRMS) (Agilent Technologies India Private Limited, Bangalore, India) was employed for the mass spectra. UV-Vis spectra were recorded in acetonitrile in the range of 200–600 nm using a Agilent Cary UV (Agilent Technologies, Santa Clara, CA, USA). An Agilent Cary 630 spectrophotometer (Agilent Technologies India Private Limited, Bangalore, India) was used to record FT-IR spectra between 400 and 4000 cm^−1^ using KBr pellets. For HPLC, the Agilent 1260 Infinity high-performance liquid chromatography system (Agilent Technologies India Private Limited, Bangalore, India), equipped with a quaternary solvent delivery system, inline degasser, autosampler, and photodiode array detector, was used. Chromatographic separation was carried out using an RP-HPLC Zorbax Extend-C18 (4.6 mm × 250 mm, 5 µm) column (Agilent Technologies India Private Limited, Bangalore, India). The mobile phase consisted of acetonitrile (25%) and water (75%) and was maintained at a flow rate of 1 mL/min. The temperature of the column was controlled at 25 °C with an injection volume of 10 µL. Detection wavelengths were set at the respective λ_max_ for each compound, and a diode array detector was employed to detect the eluted peaks between 220 and 250 nm.

### 3.3. Biological Testing

#### 3.3.1. Cell Proliferation Assay

MDA-MB-231 breast cancer cells and MCF-10A normal breast epithelial cells (from ATCC, Manassas, VA, USA) were seeded into 96-well plates at a density of 4 × 10^3^ cells per well. After overnight incubation at 37 °C, the cells were treated with varying concentrations of compound **1f** (0, 1.56, 3.125, 6.25, 12.5, 25, 50, 100, and 200 µM) and incubated for an additional 72 h. Untreated cells served as the negative control. Subsequently, 10 µL of the MTT solution (5 mg/mL) was added to each well, and the plates were incubated for 3 h to assess cell viability. The medium was then discarded, and the absorbance was measured at 570 nm using a microplate reader (Promega GloMax, Madison, WI, USA).

#### 3.3.2. Apoptotic Assay

To evaluate the pro-apoptotic effect of compound **1f**, Annexin V-FITC/PI dual staining was conducted. Cells were seeded into 12-well plates at a density of 1 × 10^5^ cells per well and treated with the compound for 24, 48, and 72 h. Post-treatment, the cells were collected using 1X trypsin, including both floating and adherent cells, and centrifuged at 4000 rpm for 5 min. The resulting pellet was resuspended in 100 µL of 1X Annexin V binding buffer and stained with Annexin V-FITC, followed by a 15 min incubation in the dark at 4 °C. Cells were then washed, resuspended in fresh binding buffer, and stained with propidium iodide before immediate analysis via flow cytometry.

#### 3.3.3. Cell Cycle Assay

Cell cycle distribution was assessed by propidium iodide (PI) staining followed by flow cytometric analysis. Cells (3 × 10^5^/well) were plated in 6-well plates and allowed to adhere for 24 h. To synchronize the cell cycle, the medium was replaced with serum-free medium for 8 h. After synchronization, cells were treated with various concentrations of compound **1f** for 24 h. Following treatment, cells were harvested with 1X trypsin, washed with cold PBS, and fixed in 70% ethanol at 4 °C for 2 h. Fixed cells were washed twice with PBS, then incubated overnight in a staining solution containing RNase A (1 µL, 10 mg/mL) and PI (5 µL, 1 mg/mL) in 1X PBS. Cell cycle distribution was subsequently determined using flow cytometry.

### 3.4. DPPH Assay

A stable free radical DPPH (2,2-Diphenyl-1-picrylhydrazyl) assay was employed to test the antioxidant activity of all the synthesized compounds. Concisely, 25 µM of the test compounds were dissolved in acetonitrile and combined with 0.1 mM of DPPH (in acetonitrile) in a 1:2 ratio (80 µL and 160 µL). Then, the mixture was left undisturbed in the dark for 30 min at room temperature. After 30 min, the absorbance of the solutions was instantly measured at 517 nm using a Varioskan Multiplate Reader (Thermo Fisher Scientific, Mumbai, India) [59]. Ascorbic acid (in acetonitrile) was used as the positive control, and andrographolide and isoandrogapholide (in acetonitrile) were used as the standards. The following equation was used to determine the DPPH radical scavenging activity:% Radical scavenging activity = (Ac − As)/Ac × 100
where Ac = the absorbance of the blank (DPPH solution), and As = the absorbance of the test samples

## 4. Conclusions

In conclusion, twenty novel 3,19-(N-phenyl-3-aryl-pyrazole) acetals of andrographolide and isoandrographolide (**1a**–**1j** and **2a**–**2j**) were synthesized and characterized. All the synthesized compounds were initially screened for their anticancer activity against 60 human cancer cell lines at the NCI, USA. Single-dose study results revealed that all 3,19-(N-phenyl-3-aryl-pyrazole) acetals of andrographolide (**1a**–**1j**) displayed superior anticancer activity compared to 3,19-(N-phenyl-3-aryl-pyrazole) acetals of isoandrographolide (**2a**–**2j**). Therefore, only compounds **1a**–**1j** were selected by the NCI for five-dose studies to find out their GI_50_s, TGI, and LC_50_s. Among the compounds tested, compound **1f** exhibited the best GI_50_s, TGI, and LC_50_s on most cell lines, followed by **1h** and **1j**, respectively. Therefore, compound **1f** was further selected for detailed in vitro analysis on normal breast cells MCF-10a and the breast cancer cell line MDA-MB-231. The cytotoxicity study of compound **1f** on MCF-10a was found to have a low inhibitory concentration. The MTT assay results demonstrated that compound **1f** exhibited strong, dose-dependent anti-proliferative activity on the cells. In addition, the apoptotic analysis revealed that compound **1f** induced apoptosis in the MDA-MB-231 cells and increased the apoptotic cells from 10% to 55% over 72 h. Furthermore, cell cycle analysis indicated that **1f** caused S phase arrest in MDA-MB-231 cells. The antioxidant activity of the synthesized compounds was evaluated using the DPPH assay. Of all the compounds tested, **1b** (R = 3-NO_2_, 60.68%) and **2b** (R = 3-NO_2_, 61.17%) demonstrated the highest radical scavenging activity. Moreover, the in silico Lipinski’s analyses of the most active compounds indicate their potential to be developed as promising therapeutic leads for cancer treatment.

## Data Availability

Data is contained within the article or Appendix A.

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
