# Peer review of "Novel 3,19-(N-Phenyl-3-(4-fluorophenyl)-pyrazole) Acetal of Andrographolide Promotes Cell Cycle Arrest and Apoptosis in MDA-MB-231 Breast Cancer Cells"

_pharmaceuticals, 2025, doi:10.3390/ph18071026_

Round 1

Reviewer 1 Report

Comments and Suggestions for Authors

This study was design to synthesize and evaluate anticancer activity of novel Andrographolide derivative, aiming to improve drug bioavailability and cancer targeted activity. Overall, the 20 novel Andrographolide derivatives was synthesis and test for their anticancer potential in various models. However there were some concern as following.

  1. In NCI screening, why the inhibition exceedung 32% was set as a criteria for active candidates? Please explain more.
  2. In Single dose study, why the study was set at 10 uM?
  3. In all cell based assay for anticancer activity, there were lacking of positive control (e.g. the commercial anticancer drug). The correlation presented in Table 2 could not replace the included a positive control drug in experiment.
  4. Does the antioxidant activity relate to anticancer activity of Andrographolide derivative? The best anticancer compound was not the same compound showed the highest anticancer activity.
  5. Drug-likeness studies for the active compounds should be compared with parent Andrographolide to indicate the their derivatives possessed the better or worst bioavailability?

  6. Anticancer activity, additionally compare with parent Andrographolide. 

Author Response

Comment-1: In NCI screening, why the inhibition exceeding 32% was set as a criteria for active candidates? Please explain more.

Our response: The 32% inhibition threshold used in the NCI-60 anticancer screening is based on preliminary criteria adopted by the National Cancer Institute (NCI) for identifying compounds with potential anticancer activity. This cutoff serves as a practical benchmark to differentiate between compounds that show minimal or no activity and those that demonstrate a biologically relevant level of growth inhibition in cancer cell lines. The 32% threshold is not absolute but is used by the NCI as a general guide to filter out inactive compounds while allowing potential leads to proceed to more detailed analyses.

Comment-2: In Single dose study, why the study was set at 10 uM?

Our response: In the NCI-60 anticancer screening, the single-dose study is typically performed at a concentration of 10 µM, which serves as a standardized protocol for the initial assessment of test compounds. This concentration is recommended by the National Cancer Institute (NCI) to ensure consistency and facilitate meaningful comparisons of cytotoxic or growth-inhibitory effects across a broad range of chemical structures tested against 60 human cancer cell lines. The choice of 10 µM provides an optimal balance; it is sufficiently high to reveal compounds with moderate to strong biological activity, yet low enough to avoid nonspecific toxicity that might occur at higher concentrations. This allows for reliable early-stage identification of promising anticancer candidates.

Comment-3: In all cell based assay for anticancer activity, there were lacking of positive control (e.g. the commercial anticancer drug). The correlation presented in Table 2 could not replace the included a positive control drug in experiment.

Our response: We included the positive control MG-132 in the 2nd plate in the cell culture experiment, but unfortunately, that plate got contaminated and we discarded the plate.

Comment-4: Does the antioxidant activity relate to anticancer activity of Andrographolide derivative? The best anticancer compound was not the same compound showed the highest anticancer activity.

Our response:  In our study, antioxidant activity was evaluated as a supplementary investigation to explore whether the Andrographolide derivatives also possessed free radical scavenging potential, which is sometimes linked to anticancer mechanisms. However, we did not intend to draw a direct correlation between antioxidant and anticancer activity in this case. Interestingly, the compound that exhibited the most potent anticancer activity was not the same as the one with the highest antioxidant activity. This suggests that, for the derivatives tested, the anticancer effects are likely governed by mechanisms independent of their antioxidant properties rather than free radical scavenging alone. Thus, while antioxidant studies were included to provide a broader biological profile of the compounds, our anticancer findings do not appear to be directly related to or dependent upon antioxidant potential.

Comment-5: Drug-likeness studies for the active compounds should be compared with parent Andrographolide to indicate the their derivatives possessed the better or worst bioavailability?

Our response: Thank you for the suggestion. Now we have compared the drug-likeness studies for the active compounds and the parent compound-andrographolide in the manuscript. (Page-15)

Comment-6: Anticancer activity, additionally compare with parent Andrographolide.

Our response:  We thank the reviewer for this valuable suggestion. However, andrographolide was not selected for further evaluation by the National Cancer Institute (NCI) following the single-dose screening, and therefore it was not included in the five-dose study. As a result, direct comparative data under the same experimental conditions are not available for the parent compound. In light of this, we have included a clarification in the revised manuscript to explain why andrographolide was not part of the five-dose study. (Page-6)

Reviewer 2 Report

Comments and Suggestions for Authors

Dear Authors,

In the abstract section, the Authors wrote: “Additionally, cell cycle analysis indicated that 1f caused G1/M phase arrest in MDA-MB-231 cells.”

The results in Table 3 did not exhibit arrest in the M phase (this should be the G2/M phase). There was an arrest in the S phase. Furthermore, after including the three independent experiments and doing the statistical analysis by calculating the P-value, 1f may be causing additional arrest in the G0/G1 phase.

Every time the Authors use an acronym, it should be defined the first time it is annotated: the GI50, TGI, and LC50.

In the apoptosis/necrosis assay, the position of the gates defining the quadrants should be identical in all the flow cytometric dot plots. However, the horizontal and vertical lines in the 0 h sample (Figure 3A) differ from the flow cytometric dot plots in Figure 3B, 3C, and 3D. Using different gate positions in this type of experiment is a big mistake. Additionally, the authors should significantly improve the compensation to separate the cell subpopulations efficiently. Also, the Authors used just one sample for each experimental condition. The controls of untreated and solvent-treated cells are missing. The authors should include results at least from three independent measurements, including standard deviation and P-values. Once the three experiments are performed, the authors should generate a bar graph including the apoptotic subpopulation (early + late) and the necrotic subpopulation. In the graph, the authors should include the standard deviations and the P-value comparing the experimental sample with solvent-treated cells; both samples should be incubated simultaneously for the same time. At the bottom of the bar graph, the authors should include some representative flow cytometric dot plots as part of the same figure.

The positions of the gates used for the cell cycle analysis via the flow cytometric histograms are incorrect and incomplete (Figure 4). The gate encompassing the G2 phase does not include the entire subpopulation; this should be the G2/M phase. The gate indicating the S phase is not included in the histograms. Moreover, the percentages for each cell cycle phase should be incorporated into the histograms. Also, the three cell cycle phases analyzed via flow cytometer should be included in the single parameter histograms, G0/G1, S, and G2/M. The position of the gates should be maintained consistent in all the histograms. The presented histograms appear to correspond to different experiment types. Moreover, in Table 3, the Authors depict a single number for each cell cycle phase. The Authors should include results at least from three independent measurements, including standard deviation and P-values.

The Authors should include the statistical analysis in each experiment.

Respectfully,

Academic Editor

MDPI Journal

Comments on the Quality of English Language

The quality of the English grammar is acceptable.

Author Response

Comment-1: In the abstract section, the Authors wrote: “Additionally, cell cycle analysis indicated that 1f caused G1/M phase arrest in MDA-MB-231 cells.”

Our response: Thank you for pointing out the oversight on our part. We have carefully reviewed the data and corrected the interpretation in the revised manuscript accordingly. (Page-1)

Comment-2: The results in Table 3 did not exhibit arrest in the M phase (this should be the G2/M phase). There was an arrest in the S phase. Furthermore, after including the three independent experiments and doing the statistical analysis by calculating the P-value, 1f may be causing additional arrest in the G0/G1 phase.

Our response: We sincerely thank the reviewer for the insightful observation and clarification. We acknowledge the correction regarding the misinterpretation of the cell cycle data.

Comment-3: Every time the Authors use an acronym, it should be defined the first time it is annotated: the GI50, TGI, and LC50. (Page-7)

Our response:  Thank you for pointing out this oversight. We have now defined the acronyms GI₅₀, TGI, and LC₅₀ at their first occurrence in the revised manuscript.

Comment-4: In the apoptosis/necrosis assay, the position of the gates defining the quadrants should be identical in all the flow cytometric dot plots. However, the horizontal and vertical lines in the 0 h sample (Figure 3A) differ from the flow cytometric dot plots in Figure 3B, 3C, and 3D. Using different gate positions in this type of experiment is a big mistake. Additionally, the authors should significantly improve the compensation to separate the cell subpopulations efficiently. Also, the Authors used just one sample for each experimental condition. The controls of untreated and solvent-treated cells are missing. The authors should include results at least from three independent measurements, including standard deviation and P-values. Once the three experiments are performed, the authors should generate a bar graph including the apoptotic subpopulation (early + late) and the necrotic subpopulation. In the graph, the authors should include the standard deviations and the P-value comparing the experimental sample with solvent-treated cells; both samples should be incubated simultaneously for the same time. At the bottom of the bar graph, the authors should include some representative flow cytometric dot plots as part of the same figure.

Our response: Thank you for the suggestion. We included results from three independent experiments and the bar graph for your perusal. We use the Flow-Jo software for FACS analysis, it is picking up the gate positions. (Page-12)

Comment-5: The positions of the gates used for the cell cycle analysis via the flow cytometric histograms are incorrect and incomplete (Figure 4). The gate encompassing the G2 phase does not include the entire subpopulation; this should be the G2/M phase. The gate indicating the S phase is not included in the histograms. Moreover, the percentages for each cell cycle phase should be incorporated into the histograms. Also, the three cell cycle phases analyzed via flow cytometer should be included in the single parameter histograms, G0/G1, S, and G2/M. The position of the gates should be maintained consistently in all the histograms. The presented histograms appear to correspond to different experiment types. Moreover, in Table 3, the Authors depict a single number for each cell cycle phase. The Authors should include results at least from three independent measurements, including standard deviation and P-values.

Our response: Thank you for the suggestion, we incorporated the percentage of cells in G0/G1, S, and G2/M in the histograms. (Page-13)

Comment-6: The Authors should include the statistical analysis in each experiment.

Our response: We included a statistical analysis for the FACS experiment.

Round 2

Reviewer 1 Report

Comments and Suggestions for Authors

Thank you for your thoughtful response to my comments and suggestions. However, there are still a few points of concern:

  • Please ensure that all responses have been properly incorporated into the manuscript. Including these explanations will enhance the clarity and understanding for researchers who are interested in this work.

Author Response

Comment 1: The Authors should include results at least from three independent measurements, including standard deviation and P-values.

Response: We thank the reviewer’s suggestion. We have now included results from at least three independent measurements for all key experiments. The data are presented with standard deviations, and P-values have been calculated to indicate statistical significance. These updates can be found in Figures 2, Figures 3a, Table 3, and Figure 4.

Comment 2: This Reviewer is unable to find the P-values in Figures 2 and 3a. Also, the standard deviation and P-values are missing in Table 3. When the P-values are included, the authors should specify what two data groups have been compared.

Response: We thank the reviewer for pointing this out. We have now revised Figures 2, Figures 3a, and Figure 4 to clearly display the P-values. In addition, we have updated Table 3 to include the standard deviations. For each reported P-value, we have specified the two data groups that were compared, as noted in the figure legend.

Reviewer 2 Report

Comments and Suggestions for Authors

Dear Authors,

Reviewer First Round:

“The Authors should include results at least from three independent measurements, including standard deviation and P-values.”

Reviewer Second Round:

This Reviewer is unable to find the P-values in Figures 2 and 3a. Also, the standard deviation and P-values are missing in Table 3. When the P-values are included, the authors should specify what two data groups have been compared.

Respectfully,

Academic Editor

MDPI Journals

Author Response

Comment-1: Please ensure that all responses have been properly incorporated into the manuscript. Including these explanations will enhance the clarity and understanding for researchers who are interested in this work.

Response: We appreciate the reviewer’s valuable feedback. All suggested revisions have now been fully incorporated into the manuscript. We have ensured that the explanations regarding statistical analyses including standard deviations, P-values, and the specific data groups compared are clearly presented in the relevant figures, tables, and legends. We believe these additions enhance the clarity and accessibility of the data for researchers interested in this work.